# Reinforcement Learning-based Decision-making for Renal Replacement Therapy in ICU-acquired AKI Patients

Haoyuan Zhang
National Institute of Health Data
Science, Peking University
Beijing, China
Advanced Institute of Information
Technology, Peking University
Hangzhou, China
School of Economics and
Management, Beihang University
Beijing, China
haoyuanzhang@buaa.edu.cn

Minqi Xiong
National Institute of Health Data
Science, Peking University
Beijing, China
mxiong5@jh.edu

Tongyue Shi
National Institute of Health Data
Science, Peking University
Beijing, China
tyshi@stu.pku.edu.cn

Wentie Liu
National Institute of Health Data
Science, Peking University
Beijing, China
wtliu@stu.pku.edu.cn

Haowei Xu
National Institute of Health Data
Science, Peking University
Beijing, China
Advanced Institute of Information
Technology, Peking University
Hangzhou, China
School of Artificial Intelligence,
Optics and Electronics, Northwestern
Polytechnic University
Xi'an, China
hwxu@nwpu.edu.cn

Huiying Zhao
Department of Critical Care Medicine,
Peking University People's Hospital
Beijing, China
zhaohuiying109@sina.com

Guilan Kong*
National Institute of Health Data
Science, Peking University
Beijing, China
Advanced Institute of Information
Technology, Peking University
Hangzhou, China
guilan.kong@hsc.pku.edu.cn

## ABSTRACT

In the Intensive Care Unit (ICU), Renal Replacement Therapy (RRT) serves as an effective tool for improving fluid balance and promoting renal function recovery in patients with severe Acute Kidney Injury (AKI). The need for RRT, the timing of its initiation and discontinuation, and the selection of its modalities require careful consideration due to its potential impact on renal function recovery and the associated risks. However, the existing Kidney Disease Improving Global Outcomes (KDIGO) guidelines provide only general recommendations, leaving room for physicians to provide subjective judgment, and thus a personalized RRT decision-making support tool is in urgent need. This study proposed to employ a value-based reinforcement learning approach to model the relationships between patient states, RRT decisions, and renal function recovery. This approach allows for action recommendations under various patient states, and can balance both short- and long-term patient outcomes. In the modelling process, patients' sequential state data was utilized, three RRT-related strategies were considered, and the reward function was defined based on the rate of estimated glomerular filtration rate (eGFR) change. The proposed reinforcement learning-based RRT decision-making model was tested using

**Corresponding Author

the AKI dataset extracted from the publicly available ICU dataset MIMIC-IV. The experimental results showed that the RRT strategy recommendations provided by our developed reinforcement learning-based decision support tool were consistent with clinical guideline and some recommendations are more rational than actual actions in specific patient cases.

## CCS CONCEPTS

• **Information systems** → **Decision support systems**; • **Computing methodologies** → **Artificial intelligence**; **Markov decision processes**; **Reinforcement learning**; • **Applied computing** → **Health informatics**.

## KEYWORDS

Reinforcement Learning, Acute Kidney Injury, Renal Replacement Therapy, Renal Function Recovery, Decision Support Tool

**ACM Reference Format:**
Haoyuan Zhang, Minqi Xiong, Tongyue Shi, Wentie Liu, Haowei Xu, Huiying Zhao, and Guilan Kong. 2024. Reinforcement Learning-based Decision-making for Renal Replacement Therapy in ICU-acquired AKI Patients. In *Proceedings of AIDSH (KDD Workshop AIDSH).* ACM, Barcelona, Spain, 6 pages.

## 1 INTRODUCTION

Value-based reinforcement learning allows for the valuation of actions during the exploration-feedback process, leading to strategy recommendation or strategy optimization [6, 9]. Although it is costly to conduct clinical experiments with real patients based on online reinforcement learning, we employed offline datasets to conduct reasonable extrapolations of reinforcement learning-based decision-making for RRT in AKI.

This study aims to develop a decision-support model that utilizes patient information available in data-rich ICU settings and recommends personalized treatment plans. The proposed model, based on the Markov Decision Process (MDP), transitions from merely estimating the probability of action implementation using existing data to calculating the action values and making informed recommendations through reinforcement learning. By associating the patient's state, action selection, and incentives, the model identifies the opportune time for RRT initiation, RRT plan change, or RRT discontinuation.

To the best of our knowledge, this is the first study to employ a value-based reinforcement learning approach to aid physicians in selecting proper RRT strategies at opportune time points for AKI patients in the ICU.

## 2 RELATED WORK

The KDIGO guideline lists the main goal of implementing RRT as maintaining homeostasis of the blood environment and allowing the kidneys to regain function [5]. In clinical practice, IHD can be used as a transition from CRRT to withdrawal. Since the KDIGO guidelines do not provide detailed guidance, there is room for clinical experts to use empirical judgment in the actual implementation of RRT.

Existing RRT-related literature [1] focuses on predicting the timing for initiation or discontinuation of RRT, comparing modalities of RRT usage, and forecasting its duration. Typically, statistical analysis and supervised learning are used, utilizing diverse patient indicator variables for prediction, and urine output is widely acknowledged as a critical indicator in studies on the discontinuation of RRT [7, 11]. Concurrently, physiological markers such as creatinine levels, urine volume, and blood electrolyte concentrations have been identified as predictive factors for the implementation of RRT [2]. However, these articles do not integrate the decision-making strategy of RRT and the prediction model into a unified framework. The proposed reinforcement learning-based models and state metrics selected for RRT initiation, discontinuation, or modality changes, introduce complexity clinical to manipulation but can take various RRT strategies into consideration from an overall perspective.

Offline reinforcement learning combines the core features of reinforcement learning with the advantages of learning from existing offline data, making it possible to construct unified RRT decision models while safeguarding clinical ethics and controlling learning costs [9]. In those applications with high trial-and-error costs, such as human-machine dialog strategies, robot control, inventory management, and automated driving, learning outcomes close to the trial-and-error effects of online interactions can be achieved by utilizing features extracted from offline datasets for model training. In addition, offline reinforcement learning has developed a set of relatively mature models and performance evaluation systems. Classical algorithms, including Policy Constraint and Regularization, not only deepen our understanding of the potential of offline reinforcement learning but also advance its practical deployment in a variety of real-world decision-making scenarios.

Reinforcement learning research based on large electronic health records (EHRs) has been conducted in various clinical contexts, such as mechanical ventilation usage, sepsis diagnosis, medication dosage control, diabetes control, and test sequence control [12]. Fatami et al. utilized the Double Deep Q Learning model to learn the effect of each discrete action on the patient's outcome (survival or death) based on the reward function's value estimation [1]. Lee et al. constructed an anesthesia ventilation control AI model for general anesthesia during the awakening period using conservative Q learning [4]. Wang et al. constructed a patient's state transfer model based on real data on diabetes control, achieved 40 levels of diabetes control using a model-based reinforcement learning approach and implemented proof-of-concept feasibility tests on real patients [10]. These studies verify the validity of their proposed models by comparing the models' recommendation results and real clinical actions in the test dataset [8], and explore the recommendation ethic and potential application value of the model, reflecting the potential of reinforcement learning in clinical areas.

Finally, the application of reinforcement learning methods to RRT scenarios for AKI in ICU is relatively rare in existing studies. One of the most relevant studies to date used a dual robust estimator to predict the timing of RRT initiation and used reinforcement learning to optimize the model parameters [2]. The above study achieved valid results in external validation, which provides confidence in the application of reinforcement learning methods in the context of RRT related decision-making.

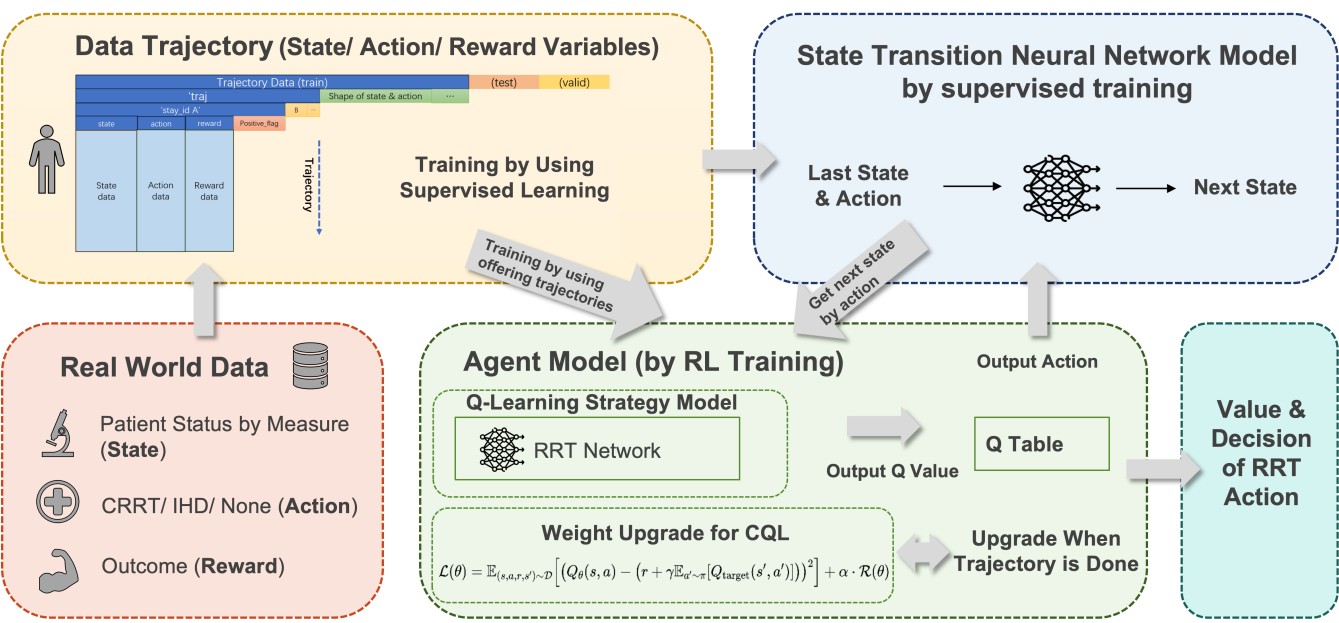

**Figure 1: Pipeline for Developing a Reinforcement Learning-Based RRT Decision-Making Tool**

## 3 METHOD

In this study, we extracted relevant temporal data from EHRs to identify the elements of actions, states, rewards, and transition probabilities, and constructed them into MDP data structure with varying sequence lengths. Utilizing state transition networks and reinforcement learning algorithms, we trained on offline dataset to balance long-term and short-term rewards through action-value estimation. The proposed model, shown in Figure 1, was subsequently deployed and validated on a test dataset to assess its efficacy and reliability in a clinical context.

### 3.1 Intensive Care Data

We utilized the Medical Information Mart for Intensive Care (MIMIC-IV) database, a large, freely available repository of de-identified health-related data from patients admitted to the critical care units of the Beth Israel Deaconess Medical Center, covering the period from 2008 to 2019 [3]. This database includes comprehensive patient demographics, time-stamped vital signs, laboratory measurements, treatments, and fluid intake data. For this study, we specifically extracted and utilized data pertaining to ICU-acquired AKI patients, which are recorded in the ICU portion of the database. The identification of AKI was based on creatinine and urine output according to KDIGO guidelines.

The process for selecting the study patients is illustrated in Figure 2. From a cohort of 42,258 patients identified with AKI during their ICU stay, we identified 2,176 patients who had accurate records of RRT, including IHD and CRRT. To support the training and testing of our model, we further refined our selection to include only those patients whose RRT treatment period is beyond 24 hours from the earliest start to the last recorded session and verified the completeness of their data on various clinical parameters. Patients

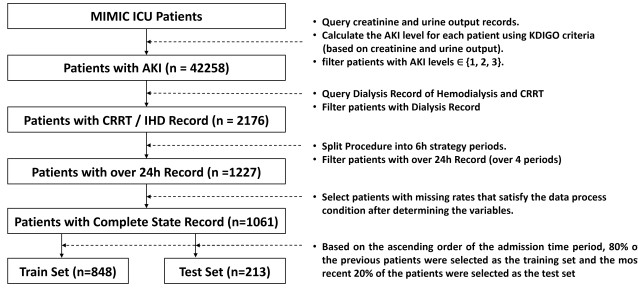

**Figure 2: Patient Population Selection Workflow**

were temporally aligned to their admission times, which were obfuscated to preserve privacy, and were then divided into training (848 patients) and testing cohorts (213 patients) based on different admission periods. For each patient included in the training and testing datasets, we retained data from ICU admission through to the outcome phase and extracted demographic data, physiological parameters, and RRT usage details for each study patient.

### 3.2 MDP Design

The MDP is a fundamental framework in reinforcement learning, composed of a finite state space S, an action space A, a reward function $r(s_t, a_t) \in R$, and a transition function $p(s_{t+1}|s_t, a_t)$. At each timestep, the agent selects an action that influences the subsequent state based on the transition probabilities and receives feedback in the form of rewards, which depend on the state and the outcomes of subsequent actions. The objective of reinforcement learning is

to learn a policy that maximizes expected cumulative rewards, enabling the agent to perform value estimation over the action space in each given state and use these estimations to recommend actions.

Based on MDP principles, we constructed an offline dataset for optimizing RRT decision-making as follows:

Firstly, we processed patient records by creating time windows. Starting from the day before the initiation of RRT in the ICU and ending at the cessation of treatment or the time point for the last ICU record, we constructed different 6-hour time windows. Fifteen variables were selected to represent the patient's state in each time window. For variables recorded multiple times within a window, we used the mean value; for missing data, we applied linear, forward, and backward interpolation to manage the gaps. In terms of actions, we focused on CRRT and IHD strategies administered in the ICU, categorizing them based on the presence and type of treatment strategy into no RRT, CRRT (indicating a more intensive patient state), and IHD (indicating a less intensive intervention). RRT actions with intervals shorter than one day were completed by backfilling, which fills the blank with previous data.

Our reward function was constructed based on the renal function recovery status, calculated by changes in the eGFR derived from creatinine levels and demographic information. In the offline dataset, we set the transition probabilities to 1, allowing for virtual exploration of different actions for each state at a single timestep.

Secondly, we formulated the specific MDP for each patient as follows:

(i) State variables: age, gender, 24 hours total urine output, blood pH, partial pressure of oxygen, calcium, bicarbonate, creatinine, sodium, potassium, sodium change rate, potassium change rate, creatinine change rat, change rate of 24 hours total urine output, and time step from start RRT.

(ii) Actions: CRRT, IHD, No-RRT

(iii) Reward function:

$$r_t = I_A(eGFR_{t+1} \geq 90) + I_A(eGFR_{t+1} < 90) \cdot \left[ \frac{2}{1 + e^{-0.5 \cdot \Delta eGFR_{t+1}}} \right] - 1$$

where the calculation of eGFR is as follows:

$$eGFR = 141 \times \min\left(\frac{Scr}{1}, 1\right)^{\alpha} \times \max\left(\frac{Scr}{1}, 1\right)^{-1.209} \times 0.993^{Age} \times \gamma$$

where $\alpha$ and $\gamma$ is defined:

- $\alpha = -0.411 + 0.082 \times I_a(\text{female})$
- $\gamma = 1 + 0.159 \times I_a(\text{is black})$

(iv) Transition function: $p(s_{t+1}|s_t, a_t) = 1$.

In our data processing approach, we shifted the reward information backward for each time window, and thus the reward associated with the current time window was derived from the following time period. This method aligns with the logical progression of renal function changes following treatment. For each series of states, we normalized the data to a standard normal distribution and capped outliers at three standard deviations ($3\delta$) to limit extreme values. This normalization process ensures that our model is not unduly influenced by extreme, non-representative data points, maintaining statistical robustness and improving the reliability of our analysis.

## 3.3 Value-based Offline Reinforcement Learning

We developed our reinforcement learning model using the Conservative Q-Learning (CQL) paradigm and updated it using the Time-Difference (TD) algorithm. The model, structured around a MDP, integrates real reward data with estimated values for training. We designed a fully connected neural network to manage state transitions, which processes state and action inputs to predict subsequent states. This network was trained on actual subsequent state data from our dataset, using supervised learning techniques.

$$Q(s_t, a_t) \leftarrow Q(s_t, a_t) + \alpha \left[ r_{t+1} + \gamma \max_{a'} Q(s_{t+1}, a') - Q(s_t, a_t) \right]$$

We employed a two-hidden-layer neural network with Adam optimizer to construct our reinforcement learning agent model by referring to a previous architecture [1]. This approach is intended to enhance the model's generalization capabilities and prevent the typical overfitting issues of offline reinforcement learning extrapolations. We incorporated L1 and L2 penalties along with a conservative Q-learning penalty for loss function design. By selecting an appropriate combination of these parameters, we aimed to balance feature learning with the generalization capabilities of the model. This methodological choice helps ensure robustness and reliability in our model's performance across varied clinical scenarios.

## 4 EXPERIMENTS

We deployed our code on a Python 3.9 and PyTorch 1.13 platform, conducting 50 training epochs for the state transition model and 400 epochs for the reinforcement learning training. From the perspective of model learning efficacy in reinforcement learning, the average loss in action valuation reflects the model's adaptation to the dataset and its capability to extract features. Figures 3(b) and 3(c) illustrate a downtrend in loss for both the state transition and reinforcement learning networks on both training and testing sets, respectively, indicating robust feature learning and generalization capabilities. Figure 3(a) displays the trend of average policy returns over iterations, depicting an overall increase in expected returns with training progression, which stabilizes, demonstrating effective learning and policy optimization by the agent, despite fluctuations due to epsilon-greedy exploration.

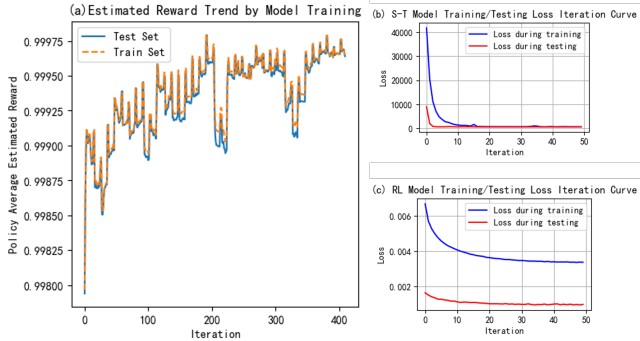

**Figure 3: Training Logs of Model**

Analyzing the model's recommendation from medical perspective is a critical step. We demonstrated the medical rationality of our decision model and showcased its application using real patient cases. On the test dataset, we employed the most comprehensively trained model to evaluate the distribution of recommended RRT strategies across different patient states. Figure 4 illustrates the frequency distribution of the model-recommended RRT actions and the patient state categorization (three groups). Figure It is a heatmap comparing actual strategies to recommended actions. Generally, the recommended actions show a higher frequency of CRRT, indicating a more conservative treatment approach, with CRRT recommendations making up more than two-thirds of the actions; recommendations for 'No RRT' are notably low, which may correlate with stable or declining rates of eGFR change in the dataset without RRT support.

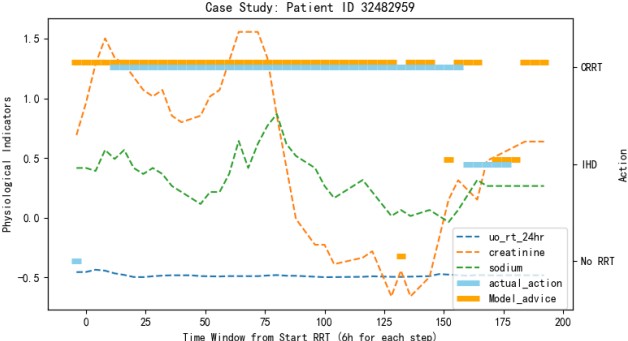

Figure 6: Specific Patient Case

rates and more CRRT for those with high inorganic salt content, reflecting a treatment preference for the latter.

Finally, we conducted a case study on patient ID 32482959 who underwent an extended duration of RRT to demonstrate our model's utility. We tracked the patient's sequence of RRT and the model's corresponding recommendations. And key renal function indicators: 24-hour urine output, creatinine, and blood sodium levels were monitored. During the initial 150-time periods, a general downward trend in creatinine concentration was observed. In the real-world setting, the patient underwent a sequence of renal replacement therapies, starting with CRRT and later transitioning to IHD. The model, consistent with the clinical course, recommended CRRT during the initial period, as shown in the Figure 6. Beyond the step 150, as the patient's creatinine levels began to rise again, IHD was administered clinically. During the time span from step 150 to 180, the model alternated its recommendations between CRRT and IHD based on varying rates of creatinine, and consistently recommended CRRT after step 180, which indicates that adherence to model-recommended CRRT could assist in better renal function recovery at later stages. This case study highlighted the model's consistency with real clinical behaviors and its ability to provide agile decision-making at critical time points.

## 5 DISCUSSION AND CONCLUSION

Utilizing the reinforcement learning framework to construct a RRT action recommendation model that aimed at renal function recovery, is a novel and challenging approach. By utilizing sequential data from patients undergoing RRT strategy, our model provides insights that are more balanced in short-term and long-term outcomes compared to cross-sectional data. Through action value estimation, we can analyze the effect and transition of actions, thereby optimizing the existing RRT strategies.

Additionally, we applied the SHapley Additive exPlanations (SHAP) methodology to analyze the interpretability of the model's variables. As shown in Figure 7, the change rate in creatinine has the strongest explanatory power within our reinforcement learning model, suggesting a rational basis for RRT action recommendation compared to traditional clinical action.

This study is uniquely positioned as one of the pioneers to apply reinforcement learning for recommending RRT strategies. While we have implemented this framework and achieved preliminary

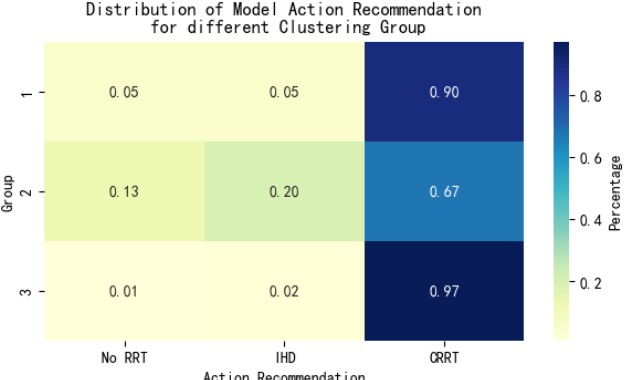

Figure 4: Heatmap of Clustering Results

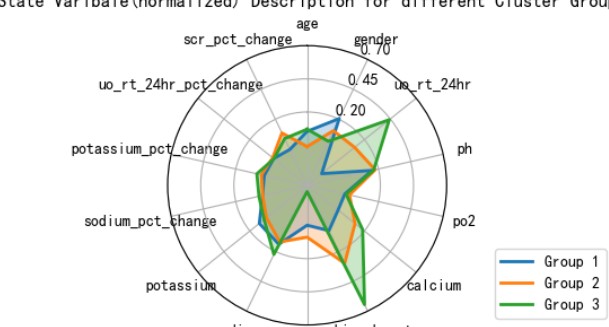

Figure 5: Distribution of Model Decision Results

From the clustering results based on patients' state variables in Figure 5, we observe the following patient characteristics: Group 1 has more male patients with lower levels of blood bicarbonate; Group 2 exhibits higher positive rates of creatinine change; Group 3 has higher blood inorganic salt content. The model tends to recommend less CRRT for patients with positive creatinine change

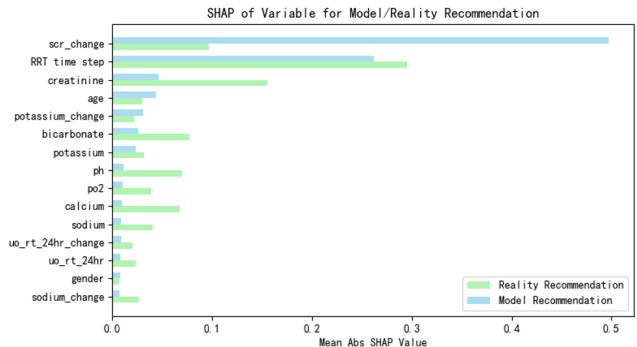

**Figure 7: SHAP Result of Model & Reality Recommendation**

results, numerous challenges and limitations await further refinement. From a clinical scenario perspective, different etiologies of AKI among patients and external medications during RRT treatment affect the predictive accuracy of the model; particularly, the use of diuretics and the physiological impact of RRT itself pose significant challenges in decision modeling. From a model perspective, the extrapolation of data features in offline reinforcement learning brings challenges to robustness, and a balanced approach between feature learning and generalization is needed. Moving forward, we anticipate more research recognizing the potential of value-based reinforcement learning in RRT domains, and hope subsequent studies will incorporate RRT dosages and other control factors in real patient cohorts to develop more guided and accurate models.

In conclusion, our reinforcement learning framework in the domain of RRT decision analysis and optimization represents an important exploratory effort. By employing MDP modeling over extended time sequences, we can evaluate actions and incorporate RRT usage adjustments into a unified decision-making framework. Results from the training and test on the MIMIC IV database indicate that our model has performed well to some extent, offering invaluable insights for RRT decision-making in the ICU.

## ACKNOWLEDGMENTS

This study was supported by the National Natural Science Foundation of China [823720951], the Zhejiang Provincial Natural Science Foundation of China [LZ22F020014], the Social Science Project of the Chinese Ministry of Education [22YJA630036], and the Beijing Natural Science Foundation [7212201].

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
