# OpenReview forum: "Reinforcement Learning-based Decision-making for Renal Replacement Therapy in ICU-acquired AKI Patients"
_KDD.org/2024/Workshop/AIDSH — KDD-AIDSH 2024 Poster_

### Official Review · Reviewer_wT3f · 2024-06-12
**A review**

**Rating:** 9
**Confidence:** 4

**Review:**

This paper presents a method that uses reinforcement learning in RRT services. The methodology is rigorous, the approach is reasonable, and the results are convincing. There are just a few minor issues:

It seems this paper is too lengthy for the workshop. Some information should be moved to the appendix in the final version.

The expression "Reinforcement learning is a cutting-edge approach" is a bit awkward. While I recognize RL's utility, it was proposed quite early. Let's use a different expression.

"RRT actions with intervals shorter than one day were completed by backfilling." The operation of backfilling is not very clear.

I hope the codes will be available and can be further studied in the workshop.

---

### Official Review · Reviewer_xtD3 · 2024-06-18
**The study is the first to apply reinforcement learning to optimize RRT decisions for AKI patients, preliminarily demonstrating the feasibility of this approach on real clinical data. However, its methodological innovation is limited, and the experimental setup lacks comprehensiveness. There is room for improvement in model details, integration of clinical knowledge, reward function design, and evaluation metrics. The most significant shortcoming is the lack of assessment on improving patient clinical outcomes and the lack of discussion on medical ethics considerations. Overall, while showing potential applications, the study needs further refinement in multiple aspects.**

**Rating:** 6
**Confidence:** 4

**Review:**

## pros
- This study is the first to introduce reinforcement learning into the optimization of RRT decisions for AKI patients. By using sequential modeling and reward function design, it constructs an end-to-end decision optimization framework, providing a new approach to addressing the challenges of selecting RRT treatment strategies in clinical practice.
- The paper is based on the MIMIC-IV and trains and tests the proposed reinforcement learning model on real clinical data from over 2,000 AKI patients. The experimental results preliminarily confirm the feasibility of using reinforcement learning to assist in RRT clinical decision-making.

## cons
- The authors selected eGFR as the basis for the reward function, but RRT decision making should not only consider the recovery of renal function. For example, the patient's quality of life, risk of complications, and consumption of medical resources are all important factors that need to be weighed. Simply using eGFR as the optimization goal may lead to "overfitting" of the strategy and overlook other clinical demands.

- The manuscript primarily uses "the consistency between model decisions and actual clinical decisions" as the evaluation metric. While this is certainly an important aspect, it cannot fully reflect the strengths and weaknesses of an RRT decision support system. More crucially, it is necessary to examine whether the clinical prognosis of patients is improved after adopting such a system.

-  The manuscript conducts an empirical study on the MIMIC-IV dataset, but the experimental setup is not comprehensive enough. The manuscript only use a single dataset and lack comparative experiments with other mainstream methods, making it difficult to highlight the advantages of the reinforcement learning approach.

-  The manuscript does not provide detailed information on the model's key parameters, such as the specific number of neurons in the hidden layers, the choice of optimizer, and the learning rate settings.

- The manuscript's methodology is primarily based on data-driven, end-to-end modeling and does not effectively integrate clinical medical domain knowledge. The authors could consider incorporating medical domain knowledge into the model.

- Although the authors selected some common clinical indicators as state variables, they lack in-depth analysis of the correlation between these indicators and RRT decision-making. Some important clinical factors may be omitted, while some redundant indicators are included in the model, which may affect the model's performance.

---

### Decision · Program_Chairs · 2024-06-28

Accept (Poster)